# The Structure of the Female Genital System of the Diving Beetle *Scarodytes halensis* (Fabricius, 1787) (Hydroporinae, Dytiscidae), and the Organization of the Spermatheca and the Spermathecal Gland Complex

**DOI:** 10.3390/insects14030282

**Published:** 2023-03-13

**Authors:** Romano Dallai, David Mercati, Pietro P. Fanciulli, Pietro Lupetti

**Affiliations:** Dipartimento Scienze della Vita, Università degli Studi di Siena, Via Aldo Moro 2, 53100 Siena, Italy; paolo.fanciulli@unisi.it (P.P.F.); pietro.lupetti@unisi.it (P.L.)

**Keywords:** female reproductive system, diving beetles, insect ultrastructure

## Abstract

**Simple Summary:**

The large group of Dytiscidae, with more than 4300 species, is characterized by an extreme variation in male and female genital apparatus, with Hydroporinae exhibiting the greatest variation. Important structures, such as the spermatheca and the spermathecal gland, show a variable configuration in the several taxa. In *Scarodytes halensis*, different from *Stictonectes optatus*, the two structures are fused in a single complex organ; each region of this complex, however, exhibits an epithelium with a quite different ultrastructure. The epithelium of the region corresponding to the spermathecal gland consists of secretory and duct-forming cells, while the region homologous to the spermatheca has a simple epithelium. *Sc. halensis* is also provided with a very long spermathecal duct connecting the bursa copulatrix to the complex organ. This duct shows an epithelium rich of microtubules and junctional structures to support the thick layer of muscles surrounding the duct. A short fertilization duct is present between the complex organ and the common oviduct. The different organization of the genital apparatus in the two species suggests the presence of a different reproductive strategy.

**Abstract:**

The fine structure of the female reproductive organs of the diving beetle *Scarodytes halensis* has been described, with particular attention to the complex organization of the spermatheca and the spermathecal gland. These organs are fused in a single structure whose epithelium is involved in a quite different activity. The secretory cells of the spermathecal gland have a large extracellular cistern with secretions; duct-forming cells, by their efferent duct, transport the secretions up to the apical cell region where they are discharged into the gland lumen. On the contrary, the spermatheca, filled with sperm, has a quite simple epithelium, apparently not involved in secretory activity. The ultrastructure of the spermatheca is almost identical to that described in a closely related species *Stictonectes optatus*. *Sc. halensis* has a long spermathecal duct connecting the bursa copulatrix to the spermatheca–spermathecal gland complex. This duct has a thick outer layer of muscle cells. Through muscle contractions, sperm can be pushed forwarding up to the complex of the two organs. A short fertilization duct allows sperm to reach the common oviduct where eggs will be fertilized. The different organization of the genital systems of *Sc. halensis* and *S. optatus* might be related to a different reproductive strategy of the two species.

## 1. Introduction

The coevolution between the sperm length and the female reproductive tract has been well determined in several insect species [1,2,3,4,5,6,7,8,9], and it was also established that the fertilization success depends on the interaction between sperm and spermatheca morphology [1].

Studies on diving beetles (Dytiscidae) have confirmed these findings and suggested that the anatomy of the female reproductive tract might have driven the evolution of the sperm features [4]. Recent works [10,11] have confirmed such a hypothesis by ultrastructural studies of the male and female reproductive system of the hydroporine *Stictonectes optatus*.

Dytiscidae female genitalic morphology has been studied by several authors [12,13,14,15,16,17,18,19]. According to Miller and Bergsten [20], female internal genitalia of diving beetle Dytiscidae are unusual among arthropod in one important aspect, the organization of the reproductive tract into a “loop” with two genital openings.

However, the configuration of female genitalia can vary within Dytiscidae, and, according to Miller [18], it can be summarized as here enlisted [18,20]:

(1) Simple genital opening, bursa copulatrix with spermathecal duct and fertilization duct both connected to spermatheca (noterid-type); (2) two genital openings, separation between bursa copulatrix and vagina, with spermathecal duct large and broad attached anteriorly to the bursa (amphizoid-type); (3) two genital openings, bursa copulatrix and vagina separated, and spermathecal duct attached anteriorly on the bursa (hydroporine-type); (4) bursa copulatrix completely reduced and spermathecal duct very slender (*Agaporomorphus*-type); (5) simple genital opening, bursa copulatrix absent and spermathecal duct and fertilization duct separated. This last condition, present among Dytiscinae, is very peculiar, as in this group, the spermatophore is placed ventral to the ovipositor in a membranous sac (the spermatophore pouch of Burmeister [15]) [18]. The marked difference between Hydroporinae and Dytiscinae may be associated with the evolution of two different mating system [20].

One of the most difficult issues still to be resolved within Hydroporinae, is the organization of the complex spermatheca–spermathecal gland. The main question is whether this complex consists of a spermatheca only, divided into two chambers or, instead, is formed by two independent organs. Angus [12] defined the spermatheca as a *diverticulum*, but Miller [18] was hesitant to accept this terminology, as he noticed that the structure could assume a great variety of forms.

We have recently described the organization of the spermatheca in the female reproductive system of the hydroporine *Stictonectes optatus* [11]. This organ consists of a spheroidal structure strictly adhering to the spermathecal gland. To sum up, the spermatheca and the spermathecal gland are functionally independent organs sharing a small tract of their epithelium.

In this work we aimed to investigate the female genital organs of another species of hydroporine, *Scarodytes halensis*, to verify whether the two mentioned organs are two separated structures, or whether they consist of a single structure, but with structurally and functionally independent regions.

## 2. Materials and Methods

Several females of *Scarodytes halensis* were collected in a small river near Grosseto (Italy) and classified by Dr. Saverio Rocchi, Museum “La Specola”, Florence.

The material was dissected under light microscopy in 0.1 M, pH 7.2 phosphate buffer to which 3% of sucrose was previously added (PB). The genital system was fixed at 4 °C overnight in 2.5% glutaraldehyde in PB, and it was observed and photographed with an Olympus stereomicroscope equipped with a Zeiss MRC5 digital camera.

After rinsing in PB, the material was post fixed in 1% OsO_4_ for 2 h, rinsed again in PB, and after alcohol dehydration (50% to 100%), it was embedded in a mixture of Epon-Araldite.

Semithin sections, obtained with a Reichert ultramicrotome, were stained with 0.5% toluidine blue and observed and photographed with a light microscope Leica DMRB equipped with a Zeiss MRC5 digital camera.

Ultrathin sections were stained with uranyl acetate and lead citrate and observed with a Philips CM10 at 80 kV.

## 3. Results

### 3.1. The General Organization of the Female Genital Apparatus

The reproductive system of *Sc. halensis* consists of two ovaries, each provided with 8 ovarioles. Each ovary continues with a short lateral oviduct that opens in a common oviduct (Figure 1a). This district receives secretions and sperm by a fertilization duct starting from the complex organ constituted by the spermatheca and the spermathecal gland. This organ, about 350 µm long, is formed by two fused globular units (Figure 1a,c). The larger unit, 145 µm wide, closer to the ovaries, corresponds to the spermathecal gland, while that one in the opposite position, 120 µm wide, is the spermatheca. The spermathecal duct opens between these globules (Figure 1a,c and Figure 2a–d). This duct, about 15–30 µm wide, extends for about 400 µm towards a structure, 450–600 µm long and 125 µm wide, consisting of a single coiled duct giving rise to a large structure of variable shape (Figure 1a). From this structure, the duct extends for about 300 µm and reaches a spherical bursa copulatrix, 75 µm wide (Figure 1a). When the long spermathecal duct is stretched, it can reach about 15 mm of length. The fertilization duct is a cuticular canal, about 30–35 µm wide, starting from the ventral side of the spermatheca–spermathecal gland complex to open into the common oviduct. This latter is a flattened district, about 250–400 µm wide consisting of a thin epithelium lined by a cuticle with finger-like cell protrusions projecting towards the oviduct lumen (Figure 2c,d).

### 3.2. The Spermathecal Duct

This long duct connects the bursa copulatrix to the complex formed by the spermatheca and the spermathecal gland (Figure 1a). The duct, as described above, after a short tract from the bursa, forms a large structure with several coils and reaches the spermatheca–spermathecal gland complex (Figure 1a and Figure 2d). The duct is 15–30 µm wide, has an epithelium of variable thickness (from 1.5 µm to 3.0 µm) and it is lined by a thick cuticle, 1.5 µm–4.0 µm high. The cuticle shows an epicuticular layer, 0.5 µm high, beneath which a 0.7 µm layer with small cavities, is present (Figure 3a,b). From the basal layer, long cuticular extensions, up to 1.2 µm long, reach the deep regions of the cytoplasm (Figure 3a,b). Numerous microtubules adhere to short densities of the plasma membrane, forming hemi-desmosomes (Figure 3a,c). Epithelial cells have polymorphic nuclei, about 5.0 µm long and 2.0 µm wide. Their cytoplasm has a large number of microtubules, among which a few mitochondria and same dense bodies are visible (Figure 3c). Beneath the basal lamina, a 0.14 µm thick layer of fibrous connective tissue is visible. This layer is in contact with the muscle fibers laying below the epithelial cells (Figure 1b and Figure 3b,c). The muscle layer consists of 13–15 cell units, each one 4.5–6.5 µm wide (Figure 1b). Hemi-desmosomes are also present at the sarcolemma level, in front of the analogous cell membrane specializations (Figure 3a,c).

### 3.3. The Spermatheca and Spermathecal Gland Complex

The two organs are fused in a single complex, about 350 µm long, showing a different epithelial structure, each one with its own specific structural organization. The region of the complex corresponding to the spermatheca has a simple epithelium, while that of the spermathecal gland has a glandular epithelium. The complex consists, as described above, of two lobes, corresponding to the spermathecal gland, the one closest to the ovary, and to the spermatheca, the distal lobe (Figure 1a,c).

The lobe corresponding to the spermathecal gland has a peripheral series of 25–30 longitudinal muscle cells, about 15.5 µm wide (Figure 1c, Figure 2b–d and Figure 4a). A basal lamina, 1.5 µm thick, rich of fibrous connective tissue (Figure 4a,b) is present between the muscle layer and a complex of secretory cells and duct-forming cells. The secretory cells are numerous and of variable shape and dimension (about 25 µm high) and are provided with large extracellular microvillated cisterns (Figure 1d, Figure 2b–d, Figure 4a–d and Figure 5a). These cisterns, 7.5–11.0 µm in diameter, show a content with heterogeneous structure, represented either by granular dense material, or by lamellate bodies of variable shape and dimension, up to 6.5 µm wide, or even by large portions of homogeneous dense and compact material. In some cisterns, long superposed straight dense structures are visible (Figure 4a–d and Figure 5a). In the cytoplasm of secretory cells, a nucleus is present (Figure 5a), it has a variable shape and size according to the available cytoplasmic space: elliptical, 12.9 µm × 5.0 µm, or roundish, 7.0 µm in diameter. In the thin cytoplasm, several mitochondria, a rich system of rough endoplasmic reticulum and a few Golgi apparatuses are visible together with dense droplets of variable dimensions (Figure 4c,d and Figure 5a).

Just beneath the microvillated apical border of a cistern, the unusual finding of a centriole structure was observed (Figure 5b). In the center of each cistern, the end-apparatus of a duct-forming cell is visible (Figure 4c,d and Figure 5c). This consists of a complex of finger-like structures, 31 nm wide, embedded in a dense material surrounding the beginning of the efferent duct of the duct-forming cell. The secretion of secretory cells reaches the apical region of the epithelial cells through the ducts of duct-forming cells. These latter cells are thin and elongated, about 4.5 µm wide in cross-section, provided with a 1.1 µm wide duct (Figure 5a and Figure 6a). Their cytoplasm shows only an elongated nucleus and few mitochondria (Figure 5a). When the ducts reach the apical epithelial cell border, they cross the cuticle (3.0–5.0 µm high) lining the epithelial cells and discharge into the gland lumen the secretion they transport (Figure 1d and Figure 6b). The lumen contains electron-dense material into which a few cross-sectioned sperm are visible.

In the posterior lobe of the complex, corresponding to the spermatheca, the epithelium has a quite different structure than that of the spermathecal gland lobe. It is simple, with cells 20–25 µm high, provided with spheroidal nuclei, 5.0–6.0 µm in diameter, placed at the basal cell region. In the cytoplasm, numerous mitochondria are visible. The epithelium is lined by a thin cuticle, about 2.0–2.5 µm high, with an epicuticular layer of 85–90 nm high. In the lumen numerous sperm are visible (Figure 1c, Figure 2a and Figure 6c–e).

### 3.4. The Fertilization Duct and the Common Oviduct

The fertilization duct of *Sc. halensis* is not easy to observe as the spermatheca–spermathecal gland complex is near to the common oviduct and the space between these organs is very reduced. By serial semi-thin sections, however, it was possible to display a cuticular epithelium lining a large cavity, about 40 µm wide, extending through the thick layer of muscle cells and connective tissue between the spermathecal gland–spermatheca complex and the common oviduct. This cavity is in continuity with a short duct, 10–12 µm wide, reaching the common oviduct (Figure 2c,d). The duct epithelium is very thin, only 3.0 µm high, and it is lined by a cuticle of variable thickness, 0.25–1.5 µm thick (Figure 7a). In the cytoplasm, a small elliptical nucleus, 3.3 µm × 1.0 µm, and few mitochondria are present. Beneath the epithelium, a series of muscle cells are visible (Figure 7a). The common oviduct is distinguishable for its irregular epithelium with finger-like cells extending towards the oviduct lumen (Figure 1c, Figure 2b–d and Figure 7b,c). Each cell hosts an elongated nucleus (6.5 µm × 2.5 µm) and some mitochondria. The epithelium is lined by a 3.5 µm thick cuticular layer. Beneath the epithelium, a layer of 6.0 µm thick muscle cells, is present (Figure 7b,c).

## 4. Discussion

As reported in the Introduction of the present work, Miller [18] and Miller and Bergsten [20] indicated that the female genital organs of Dytiscidae can be organized according to several configurations. Among these, the dytiscinae-type is characterized by the absence of the bursa copulatrix [15,18]). This group is quite particular as its members have a single genital opening, and exhibit a different strategy for the reception of the spermatophore. The Hydroporinae is the group with taxa showing a great variety of general forms and with quite important differences dealing with the size and shape of the bursa copulatrix, as well as of the spermatheca and of the spermathecal gland, the length of the fertilization and of the spermathecal ducts. A similar conclusion was also reached by Angus [12], who indicated that the Hydroporinae *Scarodytes* is characterized by a small bursa copulatrix, a long spermathecal duct and a bilobed spermatheca. The Author underlined that the two lobes of the spermatheca could be defined as “diverticulum” or spermathecal gland (receptacle sensu Miller [18]) and spermathecal, respectively.

Quite recently, a study on the female genital system of the Hydroporinae *Stictonectes optatus* [11] described the complex organization of the above-mentioned structures, giving particular emphasis to the adhesion between the spermatheca and the spermathecal gland. These two structures share a short tract of their epithelium, but maintain their own specialization. This organization has not been verified in *Scarodytes halensis*, in which the two structures are fused in a single complex, even though the epithelium of the complex appears differently specialized. As described, the complex structure consists of two lobes; those of greater size show an outer thick layer of muscle fibers and an epithelium provided with secretory cells with extracellular cisterns and duct forming cells, as occurs in many insect ectodermic glands of type 3 [21]. A long canal, produced by duct-forming cells, transports the secretions of the secretory cell up to the apical epithelium region where they are discharged into the gland lumen. Thus, this region of the complex organ has a structure very similar to that of the spermathecal gland of *S. optatus* [11]. A quite interesting finding observed in a secretory cell of the region is the presence of a centriole in the apical microvillated region of a cistern. This cell structure could indicate the initial process of the formation of a duct forming cell’s duct. Such a process, also described by Sreng and Quennedey [22] and Quennedey [21,23], starts with the production of a cilium from a centriole which then degenerates and at its place, a canal from the cistern of the secretory cells to the external surface is formed. The epithelium of the smaller lobe of the complex, that is place at the opposite side of the greater lobe, has a quite normal structure, apparently without evidence of secretory activity. The structure of this region strongly recalls the spermatheca epithelium of *S. optatus* [11]. Between these two lobes, the opening of the spermathecal duct is visible. By a comparison between the two Hydroporinae species, it is clear that *S. optatus* has a general organization different from that of *Sc. halensis*, but the ultrastructural characteristics, as well as the functions of the two regions, are quite similar in both species [11]. It can be hypothesized that the peculiar structure of the complex organ of *Sc. halensis* is a further specialization of that found in *S. optatus*, considering that these species belong to two different subtribes. This suggestion is also supported, other than the structural organization of the two organs, by the structure of the spermathecal duct, connecting the bursa copulatrix to the spermatheca. In *Sc. halensis*, this duct shows a characteristic series of thick longitudinal muscle fibers surrounding the duct epithelium, strictly anchored to the epithelial cells by an elaborated system of junctional complexes associated to microtubules, also described by Lai-Fook and Beaton [24], Noirot and Noirot-Timothée [25] and Bitsch and Bitsch [26]. This duct is extraordinarily long, up to 15 mm when it is stretched, and it coils repeatedly to form a large structure not far from the bursa copulatrix. Presumably, the contractions of such muscle fibers allow the forward progression of sperm through the spermathecal duct, up to when they have reached the spermathecal lumen. On the contrary, in *S. optatus*, this duct is short and, consequently, the series of muscle fibers is of normal appearance, with the sperm reaching the spermatheca by their own motility [11]. A long spermathecal duct, similar to that of *Sc. halensis*, is shared by other Hydroporinae [18,19] and has been described in other Dytiscidae, such as *Laccophilus maculosus* (Laccophilini), and in *Amarodytes* sp. and *Hemibidessus bifasciatus* (both Bidessini) [18].

The presence of a fertilization duct, leading the sperm from the spermatheca to the common oviduct for egg fertilization, represents the main controversial interpretation between Miller [18] and De Marzo [19]). The latter denied the presence of a fertilization duct in *Sc. halensis*, and considered the simple structure connecting the two organs as a “tendon”. As we have clearly shown in the Hydroporinae *S. optatus* [11], a fertilization duct is present, and it transports sperm up to the common oviduct. In this species, the duct is clearly visible between the two closely adherent spermatheca and spermathecal gland. On the contrary, in *Sc. halensis*, this duct is difficult to observe, as it is placed in the restricted region between the spermatheca–spermathecal gland complex and the common oviduct. Apparently, it seems a short structure, with a distal large lumen lined by a cuticular epithelium. On the other hand, the hypothesis by De Marzo [19] is unbelievable as, if it was true, sperm, at mating, would have to coming back through the long spermathecal duct to reach the side where egg fertilization should occur. The wrong interpretation by De Marzo [19] could be due to his conviction of the presence of a single female genital opening in Hydroporinae. As clearly shown by Miller [18], and how we have also described and illustrated in this and the previous work [11], two genital ducts, i.e., the spermathecal and the fertilization ducts, are evident in the female genital organs.

The presence of a spermatheca–spermathecal gland complex in *Sc. halensis* needs some further considerations. It is known in several groups of organisms that postcopulatory sexual selection generates a positive correlation between the female genital organs, and, in particular, between the spermatheca and similar sperm storage organs, and the sperm length. Evidence was obtained from *Drosophila melanogaster* [1,27,28,29,30,31]), stalk eyed flies [32,33], dung flies [34], feather-wing beetles [35], diving and scarab beetles [4,36,37], ground beetles [38], moths [39], Zoraptera [7] and Heteroptera [8,9]. Interesting, long seminal receptacles bias sperm use in favor of longer sperm; the interaction between sperm length and seminal receptacle morphology results in long sperm, being better to displace and resist displacement by competitor sperm [3].

*Sc. halensis* has a long sperm [10] and it has a very long spermathecal duct as well as a large structure, derived from the fusion of the spermatheca and the spermathecal gland. The same occurs in *S. optatus*, which also shows a long sperm and a large spermathecal structure. [11]. Although both these species share sperm conjugation with long sperm chains, only in *S. optatus* are these surrounded by a spermatostyle, a previously undescribed finding [10].

In conclusion, the two species of Hydroporinae considered here, presenting two different female genital organizations, suggest that they could perform different strategies for their reproduction. This hypothesis was also suggested by Miller and Bergsten [20]. This seems confirmed by the presence, only in *S. optatus* [11], of a device produced by males consisting in a conspicuous mating plug inserted, during mating, within the bursa copulatrix of the female to prevent remating with a second male after the former copulation. This behavior seems not to be exploited by *Sc. halensis*, which apparently is involved in a male competition at the reproduction time, without the production of plugs.

Further studies on the structural morphology of other Hydroporinae will be of great interest to establish whether the models of genital organs recently described [10,11]) are shared by other members of the subfamily. Also described by Miller [18], Hydroporinae exhibit the greatest structural variation of dytiscid female genitalia. One of the most important characters is the structure of the receptacle (sensu Miller [18]). From the studies we performed so far [10,11], it appears clear that a more extensive study on the fine structure of this organ in other members of the subfamily can lead to a better knowledge of the group.

## 5. Conclusions

The present study confirms that the hydroporine *Scarodytes halensis*, different from *Stictonectes optatus*, has the spermatheca and the spermathecal gland fused in a single complex structure. The epithelium of the complex structure, however, shows a different ultrastructure according to the region corresponding to each of the two different organs. Only that one related to the spermathecal gland has secretory and duct-forming cells. Moreover, *Sc. halensis* has a very long spermathecal duct connecting the bursa copulatrix to the complex structure. A short fertilization duct is present between the complex structure and the common oviduct. A different reproductive strategy could be performed by the two species.

## Figures and Tables

**Figure 1 insects-14-00282-f001:**
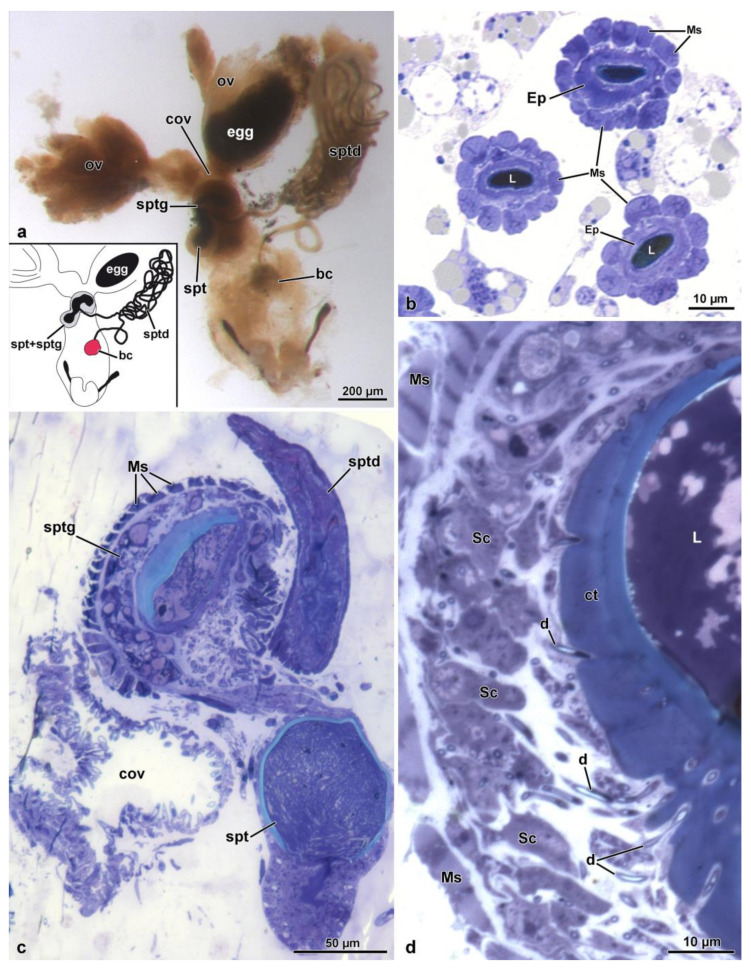
(**a**). Light microscopic view of the female reproductive organs. bc, bursa copulatrix; cov, common oviduct; ov, oviduct; spt, spermatheca; sptd, spermathecal duct; sptg, spermathecal gland. Egg at the end of the oviduct. (**b**). Semi-thin section through the spermathecal duct showing the thick layer of muscle cells (Ms) and the epithelium (Ep). L, duct lumen. (**c**). Semi-thin section of the complex spermatheca (spt)–spermathecal gland (sptg). Note the spermathecal duct (sptd) and the common oviduct (cov). (**d**). Semi-thin section of the spermathecal gland epithelia (Ep) showing the outer muscle layer (Ms), the secretory cells (Sc), the ducts (d) of the duct-forming cells, the thick cuticle (ct) and the gland lumen (L).

**Figure 2 insects-14-00282-f002:**
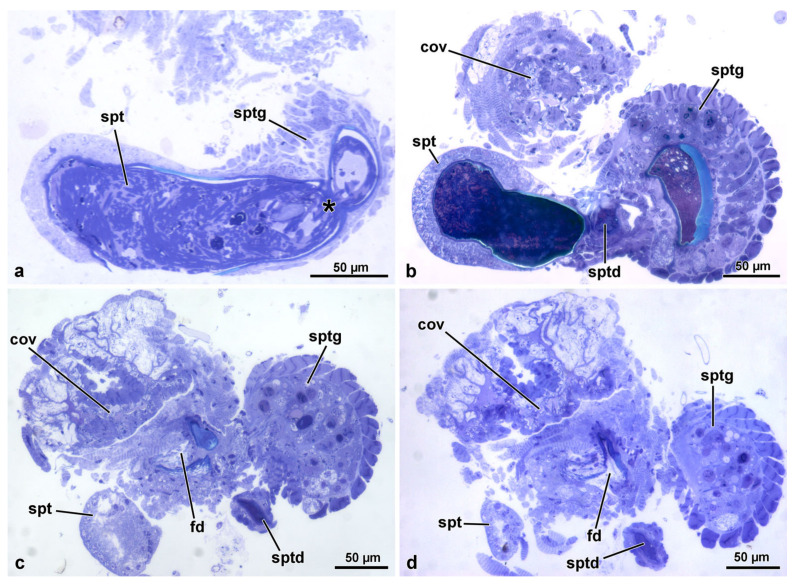
(**a**). Semi-thin section of the spermatheca (spt) and the narrow connection (asterisk) to the spermathecal gland (sptg). Note the thin wall of the spermatheca compared to that of the spermathecal gland. (**b**). Semi-thin section of the complex spermatheca (spt)–spermathecal gland (sptg). Note the quite evident different thickness of the wall of the two districts. Note also the opening the spermathecal duct (sptd). cov, common oviduct. (**c**,**d**). Consecutive semi-thin sections showing the position of the fertilization duct (fd) beneath the complex spermatheca (spt)–spermathecal gland (sptg). cov, common oviduct; sptd, spermathecal duct.

**Figure 3 insects-14-00282-f003:**
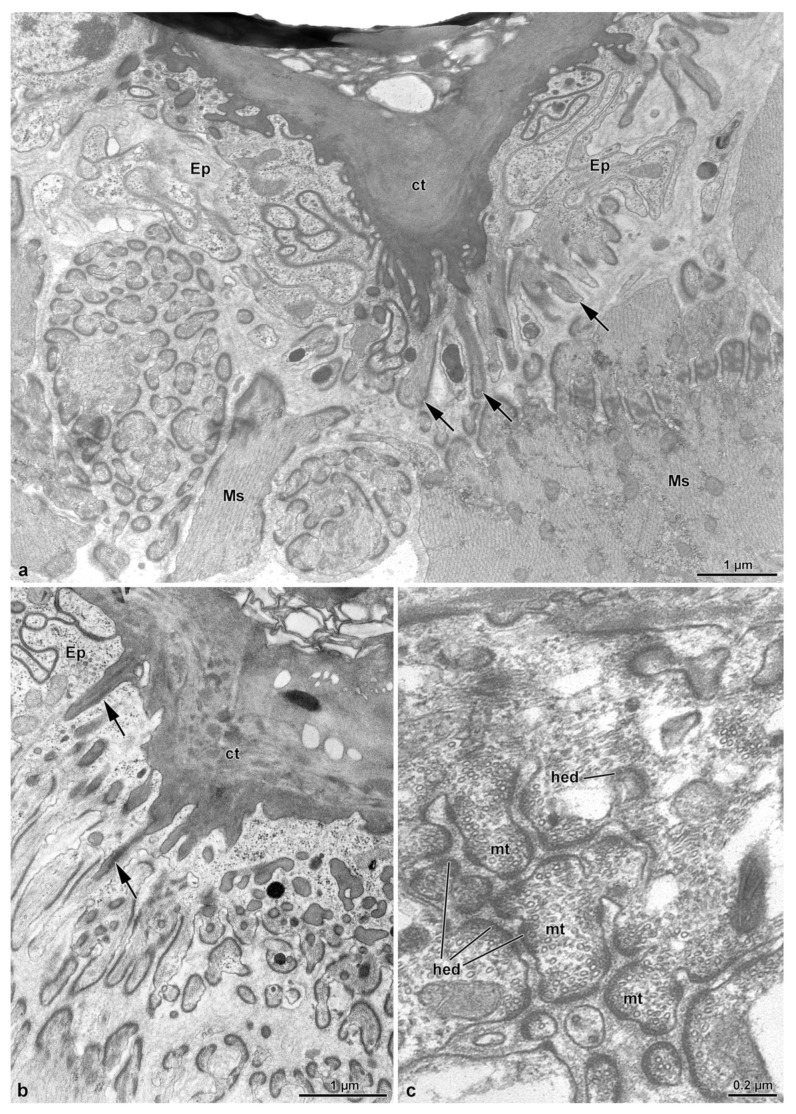
(**a**–**c**). Cross sections of the spermathecal duct. The thick cuticle (ct) lining the thin epithelium (Ep) shows long tubular prolongments (arrows) directed towards the thin cytoplasm. Around these structures, numerous longitudinal microtubules (mt) are visible reaching the basal lamina to form hemidesmosomes (hed). Beneath the epithelial cells, a thick layer of muscle cells (Ms) is present.

**Figure 4 insects-14-00282-f004:**
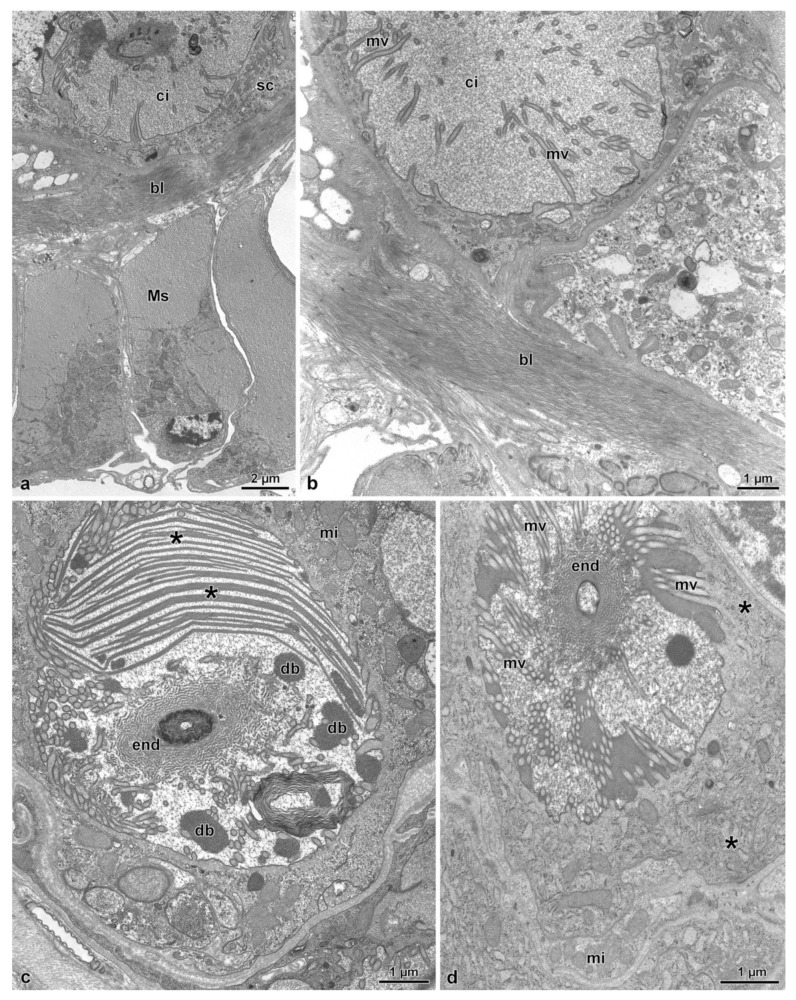
(**a**). Cross section through the basal epithelial region of the spermathecal gland. Note the secretory cell (sc) with the cistern (ci). Beneath the epithelium, a fibrous basal lamina (bl) and a thick layer of muscle cells (Ms) are visible. (**b**). Cross section of a secretory cell showing a large cistern (ci) lined by long microvilli (mv). (**c**). Cross section through a cistern of a secretory cell with the lumen filled with numerous laminar structures (asterisks) and dense bodies (db). In the center of the cistern, the “end apparatus” (end) of a duct of the duct-forming cell is visible. In the cytoplasm, mitochondria (mi) and few cisterns of rough endoplasmic reticulum are present. (**d**). Cross section through a cistern of a secretory cell. Note the long microvilli (mv) and the “end apparatus” (end). Note the mitochondria (mi) and the few cisterns of rough endoplasmic reticulum (asterisks).

**Figure 5 insects-14-00282-f005:**
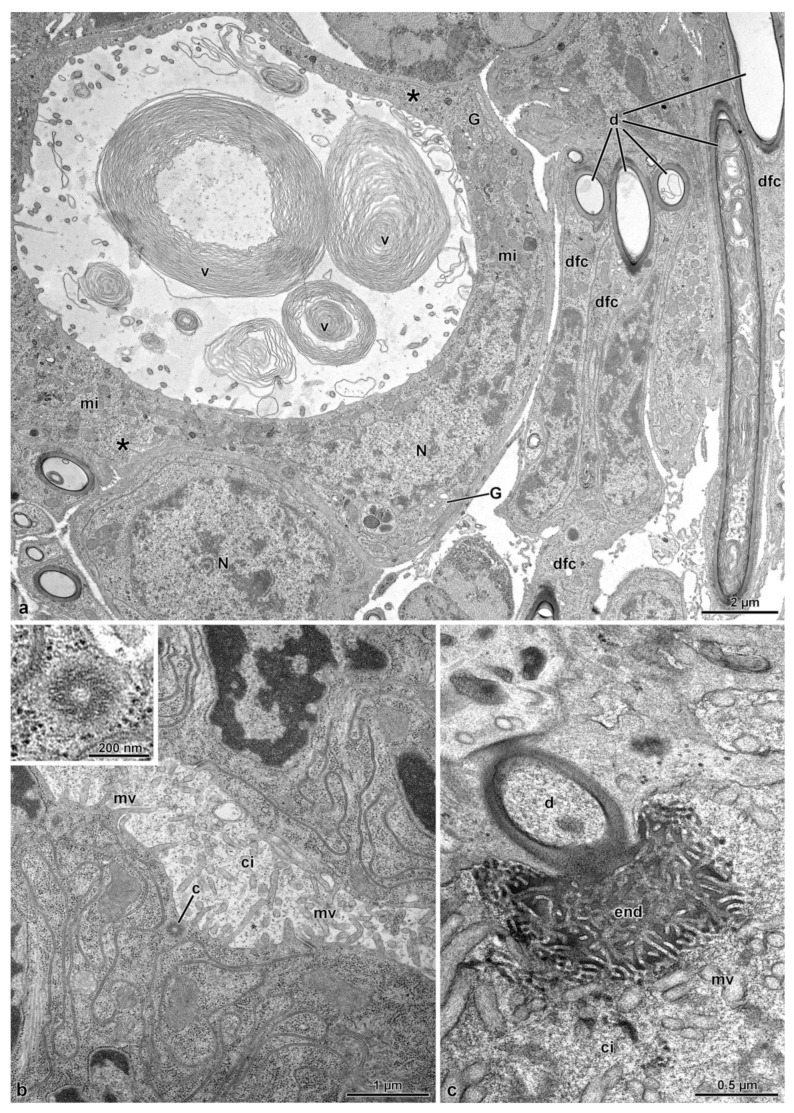
(**a**). Cross section of a large secretory cell with a cistern rich of membrane vortices (v). In the cytoplasm, an elliptical nucleus (N) is visible, and numerous mitochondria (mi), some cisterns of rough endoplasmic reticulum (asterisks) and Golgi apparatuses (G) are present. A neighboring cell shows a spherical nucleus (N). On the right side some duct-forming cells (dfc) with their ducts (d) are visible. (**b**). Cross section of a secretory cell with the cistern still to be developed. Short microvilli (mv) are visible lining the cistern. Just beneath this region, a centriole (c) is visible. In the inset, a detail of the centriole. (**c**). Cross section of the “end apparatus” (end) and of the beginning of the duct (d) of the duct-forming cell. ci, cistern; mv, microvilli.

**Figure 6 insects-14-00282-f006:**
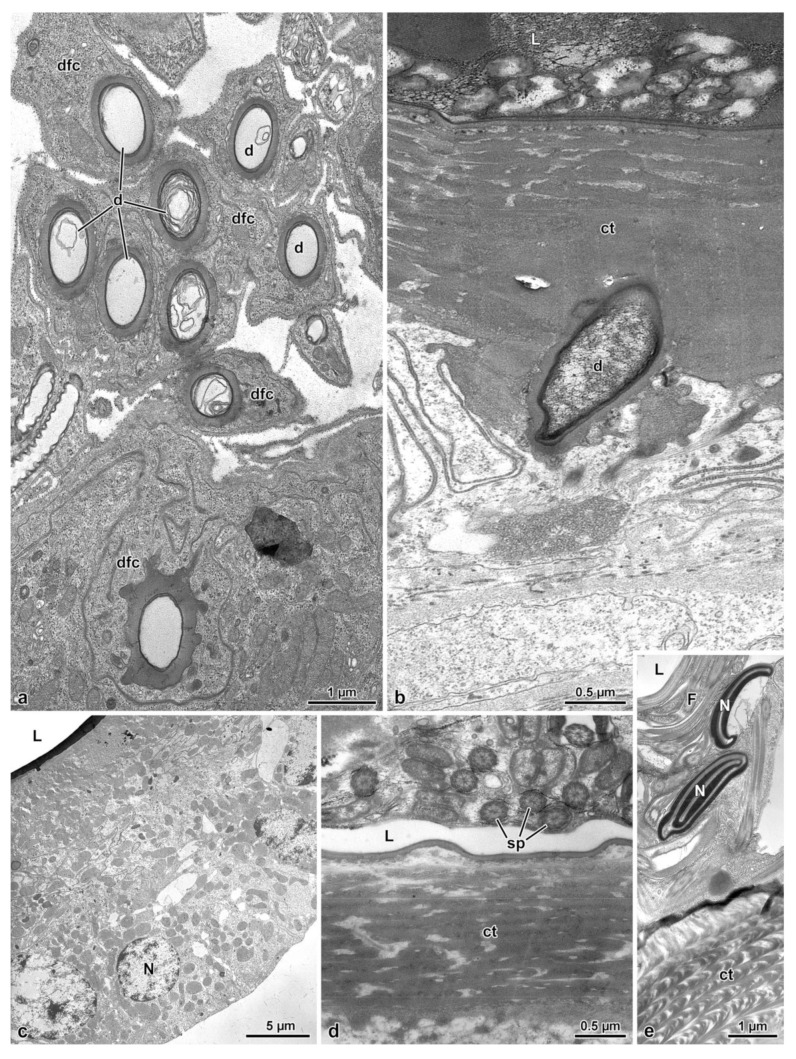
(**a**). Cross section of a complex of duct-forming cells (dfc) with their ducts (d). (**b**). Cross section of the apical region of the spermathecal gland epithelium lined by a thick cuticle (ct) that is crossed by a duct (d) of a duct-forming cell. L, gland lumen. (**c**). Cross section through the spermatheca epithelial cells. L, spermathecal lumen; N, nucleus. (**d**,**e**). Cross sections through the apical region of the spermatheca to show the lumen (L) filled with sperm cells (sp) sectioned at the flagellar (F) and nuclear (N) levels. ct, cuticle.

**Figure 7 insects-14-00282-f007:**
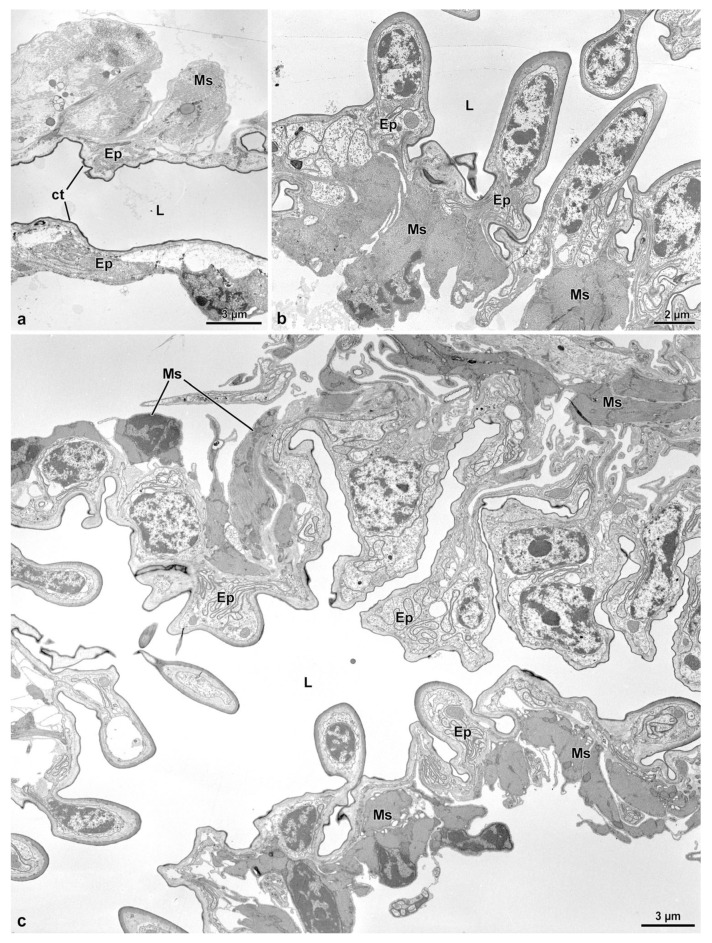
(**a**). Cross section of the fertilization duct showing a cuticular (ct) thin epithelium (Ep) with scattered muscle (Ms). L, duct lumen. (**b**,**c**). Cross sections through the characteristic epithelium (Ep) of the common oviduct, with epithelial cells often protruding into the lumen (L). Ms, Muscle cells.

## Data Availability

Data are contained within the article.

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
