# Peer review of "The Structure of the Female Genital System of the Diving Beetle *Scarodytes halensis* (Fabricius, 1787) (Hydroporinae, Dytiscidae), and the Organization of the Spermatheca and the Spermathecal Gland Complex"

_insects, 2023, doi:10.3390/insects14030282_

Round 1
Reviewer 1 Report
The authors describe a complex structure that forms part of the genital system of a water beetle. Their detailed description goes along with high quality illustrations. At some places, the text is hard to follow for a non-expert in the field of these beetles (especially lines 55-66, and 246-301).
I am confused by the authors' mentioning of the epithelium in the spermathecal gland (indicated as "ep" in figure 1d, although this figure to me shows class 3 gland cells rather than an epithelium, which in my understanding consists of a layer of similar cells that rest on a basal lamina). I do not recognize such features in the "ep" in figure 1d? My confusion grows when looking at the ultrastructural images: figure 4a says that it shows "the basal epithelial region", but again I nowhere recognize epithelial cells (nowhere in this figure is an "ep" indication)? L217 says "beneath the epithelium", but where is the epithelium...? In the Discussion, L269 mentions about an "apical epithelium", whereas the legend in figure 4 mentions about a basal epithelium. This confusion needs to be clarified!
Specific comments:
L2: species name in italics in title
L5: delete full stop at end of title
There is confusion in the description of the 2 species in the simple summary and abstract, L13 says "different from" and "different organization" in L35, which is rather contradictory to "almost identical" in L30.
L49-50: check sentence construction "...several authors; among them, (12-19)."
L51: "genitalia is unusual" (genitalia is plural, so "is" should be "are")
L55-66: too complex text to digest for a non-expert reader - better to provide schematic figure to clarify?
L94: "toluidine blue"
L99-101: what is this text doing here?
Figure 2: why have the 4 images in a vertical arrangement and not 2x2
L228: what is "a close cell"
L178: "close to the underneath common oviduct" awkward wording
L.195: "3.5 Figures" what is this text line doing here?
L.314-316: check sentence construction
Author Response
Response to Reviewer 1 Comments
Point 1: The authors describe a complex structure that forms part of the genital system of a water beetle. Their detailed description goes along with high quality illustrations. At some places, the text is hard to follow for a non-expert in the field of these beetles (especially lines 55-66, and 246-301).
I am confused by the authors' mentioning of the epithelium in the spermathecal gland (indicated as "ep" in figure 1d, although this figure to me shows class 3 gland cells rather than an epithelium, which in my understanding consists of a layer of similar cells that rest on a basal lamina). I do not recognize such features in the "ep" in figure 1d? My confusion grows when looking at the ultrastructural images: figure 4a says that it shows "the basal epithelial region", but again I nowhere recognize epithelial cells (nowhere in this figure is an "ep" indication)? L217 says "beneath the epithelium", but where is the epithelium...? In the Discussion, L269 mentions about an "apical epithelium", whereas the legend in figure 4 mentions about a basal epithelium. This confusion needs to be clarified!
The structure of the spermathecal gland region of the spermatheca-spermathecal gland complex consists of a secretory epithelium as described by Quennedey [21]. Such epithelium contains three types of cells: 1) the epithelial cells responsible of the production of the cuticle layer apically lining the epithelium, 2) the secretory cells that have a large cistern where their secretions are stored, 3) the thin duct-forming cells, able to trasport the secretions into the the gland lumen. Type 3 cells are difficult to identify because of their very reduced size. The semithin section in Fig. 1d illustrates the two main types of cells, i.e. secretory and duct-forming cells with thir respective ducts. To avoid confusion, we have removed the “Ep” labeling from the figure maintaining “Sc” and “d”.
The epithelial organization of the spermatheca-spermathecal gland complex of the species Scarodytes halensis is illustrated the series of semithin sections of Fig. 2, where the epithelium of the spermatheca and the one of the spermathecal gland are in continuity (See Fig. 2a and 2b). The epithelium of the spermatheca is simple, without any secretory cells, while the one of the spermathecal gland shows both secretory and duct-forming cells, thus resulting a secretory epithelium
Specific comments:
L2: species name in italics in title
We have corrected the specific name in italics
L5: delete full stop at end of title
We have delete the full stop ate the end of title
There is confusion in the description of the 2 species in the simple summary and abstract, L13 says "different from" and "different organization" in L35, which is rather contradictory to "almost identical" in L30.
We have corrected the sentence in the simple summary and abstract to make clear the reading
L49-50: check sentence construction "...several authors; among them, (12-19)."
We have corrected the sentence
L51: "genitalia is unusual" (genitalia is plural, so "is" should be "are")
We have corrected the sentence
L55-66: too complex text to digest for a non-expert reader - better to provide schematic figure to clarify?
We have corrected the sentence and we quoted Miller [18] who illustrated the different models of bursa copulatrix and openings within Dytiscidae
L94: "toluidine blue"
We have corrected the sentence
L99-101: what is this text doing here?
We have deleted the lines as they were erroneously left during submission
Figure 2: why have the 4 images in a vertical arrangement and not 2x2
We have rearranged the figure as request
L228: what is "a close cell"
We have corrected in “neighbouring”
L178: "close to the underneath common oviduct" awkward wording
We have rewrite the sentence
L.195: "3.5 Figures" what is this text line doing here?
We have deleted the text
L.314-316: check sentence construction
We have corrected the sentence
Reviewer 2 Report
The manuscript submitted by Dallai et al presents original observations and a detailled description of the sperm storage apparatus of an insect. The methods are electron microscopy for which this research team is well known.
The paper follows the standard organisation of anatomical descriptions, with reference to the organs, cells and ultrastructure. It is well constructed and of great interest for specialists of the anatomy of insects.
To improve the audience, I suggests some additionnal considerations on the function of the tractus, that could be facts or hypotheses.
1. What about spermatophore digestion after transfer by the male (if any spermatophore exists)? Could some structures of the bursa copulatrix being devoted to the recycling of paternal metabolic products, looking as digestive cells?
2. In the sperm reservoirs, are there glands or cells that could be dedicated to the regulation of sperm activity vs dormancy during sperm storage?
minor points:
1. what is a "relatively long sperm"? line 329
2. the beginning of results is a mistake, lines 99-101
Author Response
Response to Reviewer 2 Comments
Point 1: The manuscript submitted by Dallai et al presents original observations and a detailed description of the sperm storage apparatus of an insect. The methods are electron microscopy for which this research team is well known.
The paper follows the standard organisation of anatomical descriptions, with reference to the organs, cells and ultrastructure. It is well constructed and of great interest for specialists of the anatomy of insects.
To improve the audience, I suggest some additional considerations on the function of the tractus, that could be facts or hypotheses.
We have suggested that the long spermathecal duct of the species is the result of a positive correlation with the sperm length as it is reported in the Discussion (see Lines 345-354 ).
- What about spermatophore digestion after transfer by the male (if any spermatophore exists)? Could some structures of the bursa copulatrix being devoted to the recycling of paternal metabolic products, looking as digestive cells?
- In the sperm reservoirs, are there glands or cells that could be dedicated to the regulation of sperm activity vs dormancy during sperm storage?
Points 1 and 2: The species does not show a spermatophore. Apparently the bursa copulatrix is not involve in secretions. On the contrary, this was ascertained in the related species Stictonectes optatus (see Dallai et al. [11]).
Minor points:
- what is a "relatively long sperm"? line 329
The sperm of the species is long 0.9 mm, thus they can be considered long sperm compared to other insects sperm. We have eliminated the term “relatively”.
- the beginning of results is a mistake, lines 99-101
We have deleted the lines due to a mistake during the work submission
Round 2
Reviewer 1 Report
The authors have made the requested changes (one small typo in line 312: "could be due" instead of "could bu due"